# Brain Network Research of Skilled Shooters in the Shooting Preparation Stage under the Condition of Limited Sensory Function

**DOI:** 10.3390/brainsci12101373

**Published:** 2022-10-09

**Authors:** Feng Gu, Anmin Gong, Yi Qu, Ling Lu, Qidi Shi, Yunfa Fu

**Affiliations:** 1School of Information Engineering, Engineering University of People’s Armed Police, Xi’an 710086, China; 2School of Automation and Information Engineering, Kunming University of Science and Technology, Kunming 650032, China

**Keywords:** electroencephalogram (EEG), pistol shooting, weighted phase lag index (WPLI), brain network topology, audiovisual sensory restriction

## Abstract

Shooting is a sport dominated by psychological factors. Hence, disturbing the shooter’s sensory function during aiming will seriously affect his psychological state and shooting performance. Electroencephalograph (EEG) measurements of 30 skilled marksmen in the shooting preparation stage under noisy disturbance, weak light, and normal conditions were recorded. Therefore, the differences in neural mechanisms in the shooter’s brain during shooting aiming in different disturbance conditions were explored using an analytical approach that employs functional connectivity and brain network analysis based on graph theory. The relationship between these brain network characteristics and shooting performance was also compared. The results showed that (1) the average connection strength in the beta frequency band and connection intensity in the left and right temporal lobes of the shooters under noise disturbance were significantly higher than those under the other two conditions, and their brain networks also showed a higher global and local efficiency. In addition, (2) the functional connection intensity in the occipital region of the beta band was higher than that in the normal condition in the weak-light condition. The information interaction in the left parietal region also increased continually during the shooting process. (3) Furthermore, the shooters’ eigenvector centrality in the temporal and occipital regions with limited sensory function in the two conditions was lower than those in the normal condition. These findings suggest that noise disturbance activates the arousal level of the shooter’s brain and enhances the information processing efficiency of the brain network; however, it increases the mental workload. In weak-light conditions, shooters focus more on visual information processing during aiming and strengthen the inhibition of functions in the brain regions unrelated to shooting behavior. Audiovisual disturbance renders the cortical regions equivalent to the audiovisual perception function in the shooter’s brain less important in the entire brain network than in the normal condition. Therefore, these findings reveal the effect of audiovisual disturbance on the functional network of the cortex in the shooting preparation stage and provide a theoretical basis for further understanding the neural mechanism of the shooting process under sensory disturbances.

## 1. Introduction

An electroencephalogram (EEG) reflects the neural activity state of the brain through the understanding of the electrical activation index of the cerebral cortex [1]. Hence, in the kinematics research field, researchers provide an effective technique for monitoring the state, auxiliary assessment, and psychological training during exercise by investigating EEG characteristics [2,3]. Shooting, as a static subject, is closely related to the shooter’s psychological and physiological state and is an ideal object for EEG research [4].

The shooting preparation stage is the critical stage of shooting execution, which is also a trending topic in EEG research in kinematics. During this period, the shooter maintains a stable gun position while aiming at the target to prepare for the subsequent shooting release. A successful shooting execution requires the effective integration and coordination of specific brain regions that control visuospatial processing, planning, and motor control, therefore allowing the brain to efficiently process visual information associated with the appropriate gun positioning relative to the target to ensure optimal shooting conditions [5,6]. Hatfield et al. [7] first explored the application of EEG technology to shooting behavior. The study found that the alpha wave power of elite shooting athletes significantly increased in the left temporal and occipital regions by analyzing the EEG changes of 17 professional shooting athletes during aiming. Moreover, scholars have recently focused on exploring the changes in EEG characteristics under different experimental paradigms. Gallicchio et al. [8] analyzed the EEG signal characteristics of athletes during shooting aiming immediately before and after cycling for 3 min and found that increased heart rate load reduced the frontal midline theta power; however, it increased the temporal and occipital alpha power. Luchsinger et al. [9] also compared the shooting performance and the corresponding EEG characteristics of biathletes and cross-country skiers before and after high-intensity cross-country skiing. The study found that the frontal theta wave activity of the two athletes was significantly higher than that of cross-country skiers. However, strenuous exercise had no apparent effect on shooting performance and theta wave activity in skiers. Additionally, Woo et al. [10] employed coherence analysis to compare the interhemispheric and intrahemispheric cortical networks and visual–motor performance of 14 air-pistol athletes during practice and competition and discovered that the interhemispheric coherence during competition aiming was higher than the aiming practice process. Furthermore, Zhang et al. [11] compared the differences in EEG characteristics of 11 national archery athletes under noncompetitive and competitive shooting conditions. The study found that the power of theta, alpha, and beta waves in different brain parts was significantly higher than that in the corresponding regions under noncompetitive conditions.

Although previous scholars have confirmed the effect of different experimental paradigms on the brain’s neural mechanism during shooting aiming. These studies majorly compared the EEG characteristics of the shooting aiming period from the perspective of body load and competitive as well as ordinary conditions, and a few reports on the cognitive processing activity of the shooter’s brain during the shooting preparation stage under the limited sensory function condition. However, the impact of visual and auditory sensory disturbances (i.e., weak light and noise) on shooting performance, the differences between EEG characteristics in the shooting preparation stage under these two sensory-limited conditions compared with the normal condition, and whether these characteristics are closely linked with the corresponding shooting performance remain poorly understood.

Good shooting depends on attention given to target aiming and fine motor control [10], which results from cooperation between brain motor control and multisensory functions. Notably, the cognitive processing of sensory functions is also an important application field of EEG technology [12]. Previous EEG studies on audiovisual senses reported that the oscillations in alpha and beta bands are associated with sensory processing [13]. Fu established that the brain would have an alpha inhibition mechanism when auditory cues were employed in audiovisual stimulation [14]. Conversely, Senkowski et al. used the event-related potential (ERP) method and found that beta oscillations promoted behavior in visual, auditory, and audiovisual stimuli [15]. The effects of noise and visual deprivation on cognitive function concern many scholars in EEG-related studies with limited sensory function. Previous studies have shown that noise significantly impacts cognitive ability [16] and may be linked with behavior and cognition-related performance [17]. Noise can also cause overarousal [18], which is particularly reflected in its effect on distraction and short-term memory [19,20,21]. However, some studies presume that attention change is not obvious and even beneficial to cognitive control when the mild awakening caused by noise does not exceed the threshold [22]. In the study of visual deprivation, Noppeney (2007) discovered that long-term visual deprivation could lead to plastic changes in the visual system and the remaining intact sensory–motor system [23]. Hence, short-term visual deprivation is an established paradigm for inducing changes in neural excitability [24]. Boroojerdi et al. found a significant increase in the excitability of the visual cortex (occipital cortex) after 60 min of light deprivation [25]. However, Weisser et al. found that blinding individuals caused significant changes in neural processing of tactile modalities within 2 h, arguing that this reflects neuroplasticity caused by short-term visual deprivation [26].

The aforementioned studies confirm that EEG is a reliable technique for analyzing sensory–cognitive processing, and limitations on visual and auditory sensory functions (e.g., noise effects and visual deprivation) can alter the brain’s control over cognitive functions. Shooting is a fine motor that requires cognitive control, good cognitive perception, and motor response [27,28]. This process involves complex neural activity and requires joint analysis of information interactions between multiple functional cortices. Notably, functional interactions between cellular components distributed throughout the cerebral cortex as large-scale networks are a prerequisite for cognitive information processing [10]. The functional brain network analysis based on graph theory measures the relationship between EEG signals of different channels and regards the whole brain as a whole network. It determines the characteristics of brain functional target areas and dynamic structure changes according to local topological characteristics, such as the node degree of different nodes. It is an accurate visualization method to examine the information interaction between brain regions during cognitive processing [10,12]. They have also been applied in the field of shooting sports [27,28,29,30,31]. Therefore, this study used functional connectivity and brain network methods to analyze the brain’s neural activity during the shooting preparation stage under limited conditions of audiovisual sensory function. Notably, the crucial influence of noise disturbance and weak-light environment on the cognitive function during shooting behavior execution can be deeply understood by comparing the differences of these characteristics under different disturbance conditions. Additionally, the results of this study are projected to provide a neuroscientific reference for shooting training in complex environments (e.g., night shooting) and have a certain practical value.

The purpose of this study is to explore the functional connectivity and brain network characteristics of shooters in the pistol shooting preparation stage under noise interference and weak-light conditions by simulating an environment with limited auditory and visual functions. The shooting performance and EEG characteristics of the shooting process under noise disturbance and weak-light conditions are presumed to significantly differ from the shooting performance and brain network characteristics under normal conditions without noise and light. Nevertheless, previous studies reported that studying neural markers closely related to shooting performance can play a role in evaluating exercise levels and guiding actual training [31]. Hence, this study also explored the relationship of these characteristics with shooting performance while analyzing the differences and hypothesized that these characteristics with significant correlation with shooting performance also differed in different sensory limitations.

## 2. Materials and Methods

### 2.1. Subjects

The experimental subjects were 30 junior students (male; 21.2 ± 1.3 years old) from the Armed Police Engineering University. The students had passed the shooting course teaching and examination and mastered pistol shooting skills, regarded as the level of skilled marksman by university experts following the learning time and training intensity [32]. All subjects were right-handed, had no vision correction, had normal hearing, had no neurological or psychiatric disorders, had no major head trauma and had not undergone craniotomy. They had regular work and rest in the near future and were in a good mental state. Moreover, 24 h before the experiment, they did not take stimulant foods (e.g., alcohol, coffee, tea, or any neurological drugs) that may interfere with the experimental research. The Ethics Committee of the Armed Police Engineering University approved the experiment. All of the individuals volunteered to participate in the experiment. They understood the purpose and process of the experiment and filled in an informed consent form before the experiment. Furthermore, if discomfort occurred during the experiment, they could report it in time and apply for withdrawal.

### 2.2. Experimental Environment Setting

The indoor shooting training ground of the Armed Police Engineering University was used as the experimental site. The experimental and control groups were set up in the experiment. The noise and weak-light disturbance environments, named the noise and weak-light groups, were the experimental groups. Conversely, the environment under obvious noise conditions was used as the control group. In the noise group environment setting, the experiment selected the irregular shooting sound of bullets as the noise disturbance condition. The subjects wore Bluetooth headsets while shooting in an environment with music playing. In the volume choice, the experiment adjusted the volume from low to high to test the disturbance degree of noise to the subject. According to the participants’ description, the music volume produces a more irritable and uncomfortable mood when it exceeds 70 dB. Therefore, 70 dB was set as the volume of the noise disturbance environment in the experiment.

Alternatively, the weak-light environment was simulated with incandescent lamps with adjustable brightness in the indoor shooting training ground for the weak-light group. A GM1020 illuminance meter was used in the experiment to measure luminance. The standard of setting the weak-light environment was by causing difficulty for the subjects to achieve effective visual aiming through the alignment of the crosshair gap and the clarity of the target paper. According to the experimental method, finding the target when the illumination was 25 lx was difficult. Determining the alignment relationship of the target gap was also difficult because the target was not clear. Therefore, 25 lx was set as the illumination in the weak-light experimental environment.

### 2.3. EEG Acquisition

The EEG acquisition device was a portable 32-channel wireless EEG amplifier NSW332 produced by Neuracle Technology Co., LTD. (Zhejiang, China), which meets the mobility requirements of this experiment. The common mode rejection ratio of the amplifier was 120 dB, and the sampling frequency was set at 1000 Hz. During EEG recording, 1 Hz low-pass filtering and 40 Hz high-pass filtering were performed on the signal. Additionally, the electrode placement comprising 30 electrodes followed the international 10–20 system. The electrode positions were Fp1, Fp2, F7, F3, Fz, F4, F8, FC5, FC1, FC2, FC6, T7, A1, C3, Cz, C4, T8, A2, CP5, CP1, CP2, CP6, P7, P3, Pz, P4, P8, PO3, PO4, O1, Oz, and O2, with a ground electrode placed at the forehead. Furthermore, A1 and A2 were the reference electrodes placed at the left and right mastoid processes, with their average values set as the reference. Before the experiment, the impedance of the electrodes was adjusted and maintained below 5 kΩ. Finally, the EEGs of the resting state and the entire shooting process were collected.

The subjects sat on a chair and were relaxed during the resting EEG acquisition process. Alternatively, the subjects were not required to deliberately recall anything during the resting EEG collection. Instead, the subjects were required to keep their resting state with their eyes closed and open for 2.5 and 2.5 min, respectively.

However, in the EEG acquisition process during the entire shooting process, the subjects used the type 92 automatic pistol to shoot nonlive ammunition in a standing position under three conditions: noise disturbances, weak and bright light, and without noise. The experiment used the MSH-1 light weapons shooting training system produced by Beijing Zhongkejiecheng Technology Co., Ltd. (Beijing, China). Notably, the system used the light reflection principle to realize the aiming function of pistol shooting. Additionally, the movement track of the trace point was displayed on a computer feedback interface, and the shooting performance data of each shooter shot were recorded and stored. The subjects aimed to shoot at a shooting target paper placed 70 m away. The target paper size was 52 × 52 cm^2^, the diameter was 10 cm, the edges constantly extended outward, and the edges extended 5 cm outward for nine, eight, seven, and six rings sequentially, respectively, for the ten rings. The top edge of the target paper was five rings. The target reporter provided feedback at every interval that the athlete executed a shot on the shooting result, and the shooter adjusted the aiming point following the result. In each of the three experimental conditions, the subjects fired two sets of 30 shots at their own pace. Overall, 180 shots were executed by each subject with an interval of 10 min between each set of shots. Subjects were acclimated to the weak-light environment 8 min before the shooting commenced in the experiments under weak-light conditions. The Trigger Box is an EEG acquisition-supporting device that marks the shooting time point of the EEG signal by identifying the moment the sound of each shooting in the shooting training system appears; it was used to record the shooting time. Shooting results were recorded as 5 to 10 rounds according to the target paper (missing target was recorded as zero rounds). All subjects performed the shooting process independently, without any knowledge of each other’s performance. The subjects were advised before the experiment not to focus on their performance but on their shooting skills. Therefore, all subjects completed the same EEG signal acquisition task.

### 2.4. Signal Preprocessing

The EEG signals collected were transmitted to a computer for offline processing through the MATLAB R2014a platform. First, a finite impulse response (FIR) filter with an order of 1000 was used to perform bandpass filtering of 0.1–50 Hz on all the signals from −5 to +2 s during the aiming period (in terms of shooting time). The data were finally segmented. Considering the individual differences of the experimental subjects, the individual alpha frequency (IAF) method was used to determine the frequency band division of different subjects in the process of frequency band division. The IAF represents the frequency band between 8 and 12 Hz [33,34]. In this study, the fast Fourier transform (FFT) method was used to compute the power peak position of the occipital electrodes (O1, O2, and Oz) in the resting state with closed eyes between 8 and 12 Hz (frequency resolution, 0.5 Hz) as the IAF of the subjects. The IAF obtained determined the sub-band frequency of each subject as follows: the theta, alpha, and beta1 frequency bands were defined as IAF-6-IAF-3, IAF-2~IAF+2, and IAF+3–230 Hz, respectively. According to the divided frequency bands above, FIR bandpass filters of theta, alpha, and beta frequency bands were constructed to filter all the EEG signals of the subjects; the filtered EEG signals in a specific frequency band were also obtained [35]. In terms of time division, the EEG signal was intercepted from 3 s before the shooting to the time of shooting as the EEG data during the shooting aiming period, recorded as one trial because the subjects can usually complete a shot every 5 s on average. Finally, the EEGLAB toolbox was used to eliminate the EEG tests affected by artifacts. Therefore, 5195 available trails were obtained for 30 subjects during aiming under the three environments, with an average remaining 57 trails per subject (removal rate ≈3.8%).

The experimental analysis process was shown in Figure 1. The EEG data format of subjects under three different sensory limitations was 30 × 3000 × 57 × 3 × 30 after preprocessing, representing 30 channels, 3000 data sampling sites, 57 trials, 3 analysis bands, and 30 subjects.

### 2.5. EEG Functional Connectivity Based on WPLI

This study used WPLI to evaluate the functional connectivity between EEG signals. WPLI measures the connectivity between two neural signals by calculating the phase difference of EEG in different channels [36]. It is an improved measurement method for phase synchronization of electrophysiological signals in response to noise and volume conduction effects. It weighs the phase lead and lags following the size of the cross-spectrum imaginary part to eliminate the zero-lag phase difference effect, thereby reducing the volume conduction effect to a considerable extent and improving the signal-to-noise ratio [37]. Furthermore, the WPLI method is also suitable for real-time monitoring and functional connectivity calculation due to its low computational complexity and robust statistical power in phase detection. The formula for calculating WPLI values between EEG signals of different leads is as follows:(1)$${\mathrm{WPLI}}_{\mathrm{xy}}=\frac{\left|\langle \left|ℑ\left(S_{\mathrm{xy}}\left(t\right)\right)\right|\mathrm{sign}\left(ℑ\left(S_{\mathrm{xy}}\left(t\right)\right)\right)\rangle \right|}{\langle \left|ℑ\left(S_{\mathrm{xy}}\left(t\right)\right)\right|\rangle }$$

$S_{\mathrm{xy}}\left(t\right)$ represents the cross-spectrum of EEG signals *x*(*t*) and *y*(*t*), ℑ(•) represents the imaginary part, which is the analytic signal obtained by Hilbert transformation of narrowband signals in each frequency band and removing 10% of the data at both ends of the analytic signal, and 〈•〉 represents the average value for a given time. The value of WPLI ranges from 0 to 1. The higher the WPLI value, the higher the coupling degree of oscillatory neural activity. The data format of the WPLI connection matrix for each perception-constrained environment was 30 × 30 × 3 × 30, representing the 30 × 30 WPLI connection matrix for each group, three analysis bands, and 30 subjects by averaging the WPLI connection matrix across all EEG channels.

Additionally, this paper used the event-related weighted phase lag index (ERWPLI) as an indicator to quantify the rate change of functional connection strength to study the neural mechanism changes of shooters in the shooting preparation stage [38]. ERWPLI describes the degree of changes in the brain connection strength of subjects during aiming. In this experiment, the average connection strength of subjects in the span test times within −4~−3 s before shooting and launching was taken as the baseline value during the aiming period. ERWPLI is calculated as follows:(2)$$\mathrm{ER}WPLI(f)=\frac{WPLI(f)-\bar{R(f)}}{\bar{R(f)}}$$
where WPLI(f) represents the functional connection strength at a frequency band based on WPLI. R(f) represents the functional connection strength of each frequency band in the baseline period. This paper calculated the functional connection strength change rate in three analysis bands. The calculation described above was performed using the MATLAB R2014a platform.

### 2.6. Functional Brain Network Characteristics during Aiming

This experiment used the brain network analysis method based on graph theory to analyze the topological characteristics of the brain network in the WPLI connection matrix of the subjects in different shooting conditions. The analysis procedure was used to transform the functional connection matrix obtained by the WPLI method into the adjacency weight matrix. Different brain regions were regarded as nodes and connections as edges using the graph theory method to extract the topological characteristics of each subject’s brain network in each analysis band [31].

The threshold in this experiment was not set. However, the connection values between all nodes were retained to form a functional weighted brain network. Therefore, the topological characteristics of the brain network from a global and local perspective were jointly analyzed. The average clustering coefficient, characteristic path length, and global efficiency of the network were selected to analyze the global topological characteristics. Alternatively, the eigenvector centrality and local efficiency were also selected among the local topological characteristics. The calculation process above was implemented using the Brain Connectivity Toolbox, Vander Bilt Department of Biomedical Engineering, Vanderbilt University, Nashville, TN, USA [39].

### 2.7. Statistical Analysis

This study assumes significant differences in the brain network connections and topological characteristics of shooters during the shooting preparation stage in the noise disturbance environment and weak-light conditions compared with the normal condition. The Kolmogorov–Smirnov test (K-S test) was initially used to evaluate the characteristics of brain networks to test the hypothesis, and the test results showed that not all the data conformed with a normal distribution (*p* < 0.05). Therefore, a nonparametric Wilcoxon rank-sum test was used to evaluate the difference between these connection values and characteristics [40]. Moreover, 0.05 and 0.01 was set as the test level with significant and highly significant differences in sample data, respectively. Finally, the false discovery rate (FDR) method was adopted to correct the statistical results [41].

Concurrently, finding the differences in neural markers closely related to the strength, rate of change, and network topology of functional connections in different perception-constrained environments with their average performance of 60 shots were also attempted. Additionally, after analyzing the location of the brain regions where the statistically significant correlations occurred, which functional connections and topological characteristics between brain regions had significant positive or negative effects on shooting performance under different sensory limitations were further analyzed. Furthermore, the K-S test on the average scores of 60 shots of each subject under two perceptual constraints was conducted to verify this hypothesis. The average scores were found to not follow the normal distribution (*p* < 0.05). Therefore, the nonparametric Spearman rank correlation test was selected for correlation analysis, and the correlation coefficient (*r*) was calculated [31,42]. Similarly, the FDR was used for multiple inspections and corrections of *p* values. The statistical analysis was also performed with the statistical test toolbox of the MATLAB R2014a platform.

## 3. Result

### 3.1. Shooting Performance Differences

Figure 2a shows that the average ring values of the control, noise, and weak-light groups were 8.02 (±0.76), 8.00 (±0.81), and 5.38 (±1.56), respectively, from the shooting performance indicators. The Wilcoxon rank-sum test was used to evaluate the shooting performance of the subjects under the three conditions. The results showed that no significant difference was observed between the noise and control groups (*p* > 0.05). Conversely, the shooting performance of the weak-light group was significantly lower than that of the control group (*p* = 3.787 × 10^−6^) and the noise group (*p* = 4.225 × 10^−6^).

### 3.2. Functional Connectivity Differences during Aiming

The experiment calculated the phase synchronization of the subjects in the three frequency bands (theta, alpha, and beta) under three different conditions. Figure 2b shows the mean WPLI value obtained by averaging the WPLI connection matrix of all subjects in each group in the channel and subject dimensions. The mean WPLI value of the noise group was significantly higher than that of the control (*p* = 0.0015) and the weak-light (*p* = 0.0439) groups in the beta frequency band. Conversely, no significant difference was observed between the mean WPLI values in the theta and alpha bands.

The upper brain topographic map, which depicts WPLI with significant differences (*p* < 0.05) between the functional connectivity of the brain during aiming between the noise and the weak-light groups compared with the control group, is illustrated in Figure 2c,d. The results showed that in the theta band, the WPLI between more electrodes in the control group was significantly higher than in the noise and the weak-light groups. Conversely, in the alpha band, most connection values of the noise group were higher than those of the control group, and the functional connection differences among electrodes Fz -T7, FC5-FC2, FC5-Cz, C3-Cz, F4-P4, and C4-P4 were very significant (*p* < 0.001). Additionally, in the weak-light group, the WPLI between most electrodes was lower than that in the control group, with very significant differences in WPLI between electrodes Fp2-Cz, F7-F8, F7-P4, Cz-P7, and C3-O1 (*p* < 0.001). In the beta frequency band, both the noise and the weak-light groups showed significantly more connections than the control group. Notably, WPLI with significant differences between the noise and control groups was majorly concentrated in the right prefrontal, central, and left parietal regions. The difference in functional connections between the weak-light and control groups was more significant in the right hemisphere, where the connections with very significant differences (*p* < 0.001) were Fp2-O2, Fz-O2, F8-P4, FC2-CP2, FC6-CP6, Cz-C4, and P4-P8.

Furthermore, to more accurately study the electrode position of the significant connection value under sensory restriction, the row/column directions of the WPLI connection matrix of all subjects in the experimental and control groups were averaged to calculate the average WPLI on the different channels. The Wilcoxon rank-sum test was then used to obtain the results of significant differences in the average connection strength on each channel, as shown at the bottom of the brain topographic map (Figure 2c,d). The black node in the figure represents the position of the electrode. First, in the theta band, the node with the most significant difference in the average WPLI between the noise and control groups was in the left frontal region (Fp1: *p* = 0.0036; FC5: *p* = 0.0272), and both were higher in the control group. The mean WPLI in the right occipital region (O1: *p* = 0.0256) in the weak-light group was significantly higher than in the control group. Second, in the alpha frequency band, the mean WPLI of the noise group in the central region (Cz: *p* = 0.0088; C3: *p* = 0.0020; and C4: *p* = 0.0202), the left temporal region (T7: *p* = 0.0403), the right temporal region (T8: *p* = 0.0342), and the parietal region (P3: *p* = 0.0305; P4: *p* = 0.0072) was higher than that of the control group. The mean WPLI in the weak-light group was significantly lower in the left frontal region (F7: *p* = 0.0021) than in the control group. Last, the mean WPLI of the noise group in the beta frequency band showed significant differences in most regions of the entire brain, showing very significant differences at nodes FC2 (*p* = 0.0029), Cz (*p* = 0.0033), C3 (*p* = 0.0083), CP1 (*p* = 0.0059), P3 (*p* = 0.0023), and PO3 (*p* = 0.0036). Moreover, the weak-light group showed significant differences only in the occipital region (Oz: *p* = 0.0477; O2: *p* = 0.0268).

### 3.3. Differences in Functional Connectivity Changes

The functional connectivity change rate was also a focus of this study. Figure 3a depicts the mean and variance of mean WPLI values for skilled shooters in the baseline period (−4 to −3 s before firing) and during aiming in the three conditions. The mean WPLI values in the three conditions of the theta band were found to be higher than that in the baseline period (*p* < 0.05) based on the results presented (Figure 3a). However, it was not significant in the other two frequency bands.

ERWPLI reflects the change degree of functional connection strength during aiming. The test results of ERWPLI between the noise and weak-light groups compared with the control group in three frequency bands are shown in Figure 3b,c. Overall, the noise group had more ERWPLI than the control group in the theta and beta bands. However, most brain regions in the alpha band of the weak-light group showed higher WPLI than the control group. The brain map at the bottom of the figure shows significant differences in the average ERWPLI after averaging the row/column directions of the functional connectivity change rate matrix. These results were consistent with the significant connections in the upper figure. Furthermore, in the noise group, the mean ERWPLI at nodes F3 (*p* = 0.0272), FC5 (*p* = 0.0072), Cz (*p* = 0.0362), C4 (*p* = 0.0190), CP6 (*p* = 0.0202), Pz (*p* = 0.0083), P4 (*p* = 0.0008), and P8 (*p* = 0.0342) in the theta band, and nodes F8 (*p* = 0.0305), FC1 (*p* = 0.0427), and T8 (*p* = 0.0382) in the beta band were significantly lower than those in the control group. Moreover, the weak-light group had significant differences in the alpha band, with higher mean ERWPLI adjacent to the left parietal region (CP5, *p* = 0.0041; P5, *p* = 0.0063; and P7, *p* = 0.0179).

### 3.4. Differences in Topological Characteristics of Brain Networks

Figure 4a describes the mean, standard deviation, and statistical test results of the brain network global topological characteristics in the experimental and control groups’ theta, alpha, and beta bands. The theta band showed no significant difference. In the alpha band, the three global topological characteristics of the noise and weak-light groups showed significant differences. The noise group’s average clustering coefficient and global efficiency were higher than those of the weak-light group (average clustering coefficient, *p* = 0.0404; global efficiency, *p* = 0.0481), but the characteristic path length was the opposite (*p* = 0.0382). Additionally, in the beta band, the global efficiency of the noise group was higher than that of the control (*p* = 0.0008) and weak-light (*p* = 0.0426) groups. The characteristic path length of the noise group was lower than that of the control (*p* = 0.0015) and weak-light (*p* = 0.0483) groups. Meanwhile, the average clustering coefficient of the noise group was higher than that of the control group (*p* = 0.0020).

Figure 4b,c shows the significant differences between the local efficiency of the experimental and control groups. The colored regions in the brain topographic map represent the brain regions with significant differences. The red indicates a higher topological characteristic value of the experimental group than that of the control group, and the blue is vice versa. Comparing the noise and the control groups revealed significant differences in the three frequency bands, and that of the noise group was higher. The significant differences in the theta band were found at node Fp2 (*p* = 0.0094) and node FC5 (*p* = 0.0404). The nodes with significant differences in the alpha frequency band were mostly concentrated at the connecting line of the left and right temporal regions, where nodes FC5 (*p* = 0.0202), Cz (*p* = 0.0139), C4 (*p* = 0.0115), T7 (*p* = 0.0362), T8 (*p* = 0.0228), CP2 (*p* = 0.0256), CP6 (*p* = 0.0476), P3 (*p* = 0.0256), and P4 (*p* = 0.0131) had significant differences, whereas node C3 had a very significant difference (*p* = 0.0021). Notably, most nodes in the beta band showed significant differences. The nodes with very significant differences were Fp1 (*p* = 0.0025), Fz (*p* = 0.0051), F4 (*p* = 0.0077), FC2 (*p* = 0.0023), Cz (*p* = 0.0008), C3 (*p* = 0.0033), C4 (*p* = 0.0059), CP1 (*p* = 0.0038), CP6 (*p* = 0.0094), P3 (*p* = 0.0004), and PO3 (*p* = 0.0003). However, a significant difference was found only in the alpha frequency band in the weak-light group. The test results showed that the local efficiency of the weak-light group at node F7 was significantly lower than that of the control group (*p* = 0.0038).

Significant nodes exist in the three frequency bands in the statistical test of the eigenvector centrality. Figure 4c shows the comparison results between the noise and control groups, indicating the nodes with significant differences in the three frequency bands. First, the eigenvectors centrality of the noise group at node Fp1 was high (*p* = 0.0007), whereas that at node T7 was lower than that of the control group (*p* = 0.0476) in the theta band. Second, in the alpha band, the noise group had higher eigenvector centrality at Cz than the control group (p = 0.0063). Third, the eigenvector centrality at node T8 of the noise group was significantly lower than that of the control group (*p* = 0.0256) in the beta band. In comparing the weak-light and control groups, the eigenvector centrality at node Fp1 was significantly higher than that of the control group at the theta band (*p* = 0.0101). Conversely, the control group was higher at node O1 (*p* = 0.0158). However, the nodes with significant differences in the alpha band in the weak-light group were lower than those in the control group (F7: *p* = 0.0055; Pz: *p* = 0.0215; P4: *p* = 0.0079; and Oz: *p* = 0.0404). Furthermore, the eigenvector centrality in the weak-light group was higher than that in the control group at the right frontal region in the beta band (F3: *p* = 0.0427), whereas it was lower near the parietal, occipital region (Pz: *p* = 0.0202; PO4: *p* = 0.0088; and Oz: *p* = 0.0158).

### 3.5. Correlation between Brain Network Characteristics and Shooting Performance

No significant correlation was found between the mean WPLI values of the experimental and control group in the three frequency bands during correlation analysis between the brain network characteristics and shooting performance. Figure 5a shows the significant correlation between WPLI and the corresponding shooting performance of skilled shooters while aiming under sensory constraints. Overall, more significant functional connections were noted between the weak-light group and the shooting performance. Significant connections in theta band showed a more negative correlation. In the alpha band, the noise group showed a more positive correlation in the parietal and occipital regions of the left hemisphere, whereas the right hemisphere showed a negative correlation. The weak-light group showed more significantly positive connections on most nodes of the entire brain compared with the noise group. Additionally, most of the WPLI between electrodes in the two groups with significant shooting performance showed a positive correlation in the beta band.

Figure 5b shows the significant connections between the ERWPLI and shooting performance of the noise and weak-light groups following the Spearman rank correlation test. The figure showed that the noise group had more negative correlation connections with the shooting performance in the theta band. In the alpha band, the two groups showed opposite correlations in the left and right hemispheres. Both had more significant connections in the left hemispheres that were positively correlated with shooting performance. Most of the right hemispheres had significant connections that were negatively correlated with the shooting performance. Notably, significant connections between the noise group and shooting performance were mostly concentrated in the parietal region, whereas the weak-light group was mostly concentrated in the frontal region. The ERWPLI of both groups in the theta band showed a more positive correlation with firing performance.

In analyzing the global topological characteristics, no significant correlation was found between global efficiency, average clustering coefficient, eigenvector centrality, and shooting performance. Figure 5c shows the nodes in the local topological characteristics that had a significant correlation with the shooting performance. In the local efficiency, the nodes with a significant correlation with the shooting performance from the noise group could not be found. However, a significant negative correlation was observed between the weak-light group and shooting performance at node O1 in the theta band (*r* = −0.4013, *p* = 0.0343). Additionally, the local efficiency at nodes Fp2 (*r* = 0.3843, *p* = 0.0435), F4 (*r* = 0.4648, *p* = 0.0127), CP1 (*r* = 0.4962, *p* = 0.0072), Pz (*r* = 0.4445, *p* = 0.0178), O1 (*r* = 0.3870, *p* = 0.0419), and O2 (*r* = 0.4073, *p* = 0.0331) in the alpha band had a significant positive correlation with the shooting performance. Furthermore, a significant positive correlation was found in the beta band between the local efficiency and shooting performance in the right frontal region (Fp2: *r* = 0.4472, *p* = 0.0176), the right parietal region (CP6: *r* = 0. 3818, *p* = 0.0456; P8: *r* = 0.4081, *p* = 0.0311), and the left parietal region (P3: *r* = 0.5272, *p* = 0.0039; P7: *r* = 0.4951, *p* = 0.0070) in the weak-light group.

Figure 5d shows the nodes with a significant correlation between the eigenvector centrality of the noise and the weak-light groups and the shooting performance. Significant negative correlations were found in the theta band at nodes P8 (*r* = −0.5823, *p* = 0.0007) and O1 (*r* = −0.4910, *p* = 0.0080) in the noise and weak-light groups, respectively. In the alpha band, the noise group positively correlated with the firing performance at node O1 (*r* = 0.4370, *p* = 0.0096). However, a significant negative correlation between the eigenvector centrality was found at nodes Fz (*r* = −0.4230, *p* = 0.0249) and T8 (*r* = −0.3926, *p* = 0.0388) and shooting performance in the weak-light group (*p* = 0.0388). Simultaneously, a positive correlation was found between shooting performance at C3 (*r* = −0.4008, *p* = 0.0345). In the beta band, the noise group had a significant negative correlation at node CP5 (*r* = −0.4374, *p* = 0.0109), while the weak-light group showed a significant negative correlation with the shooting performance at node CP6 (*r* = −0.4374, *p* = 0.0011) and a significant positive correlation with the firing performance at nodes P7 (*r* = 0.3944, *p* = 0.0378).

## 4. Discussion

This study compared the differences between the brain network characteristics of skilled shooters in the shooting preparation stage during normal and limited audiovisual conditions to explore the effects of different disturbance conditions on the brain function network of shooters during aiming.

### 4.1. Analysis of Functional Connection Difference during Aiming

Brain connectivity is important for explaining how neurons and neural networks process information [43]. Abrams et al. assume that the efficient communication of information between functionally specific brain regions is necessary for good cognitive processing [44]. This study tests the differences in the brain’s functional connectivity of skilled shooters under three conditions. Overall, only the difference in the beta band showed significance in evaluating the mean WPLI value under the three frequency bands. Additionally, the mean WPLI value of the shooter’s brain under noise disturbance during aiming was significantly higher than that of the other two groups in this frequency band. Previous studies have suggested that beta rhythm impacts cognitive processes (e.g., attention, working memory, and audiovisual integration), participates in sensorimotor integration, and is linked with increased arousal levels [12,45]. Meanwhile, higher connection strength represents higher energy consumption [31]. This result shows that the shooters’ brains under noise disturbance conditions have higher arousal degrees, stronger information interaction ability, and more energy consumption than in normal and weak-light environments. Furthermore, previous studies on the noise effect reported that noise could stimulate the brain and increase the mental load [46] and surmised that the mild brain stimulation caused by noise causes less distraction and is conducive to cognitive control [22]. Combined with the result that showed no significant difference between the shooting performance of the noise and the control groups, exerting a certain degree of noise disturbance during aiming is believed to affect the shooting performance of the shooter minimally and can enhance brain arousal but would cause a mental load increase.

Information communication between different cerebral cortexes reflects the subjects’ level of consciousness, the ability of the brain to integrate information, and the ability of cognitive processing [43]. In the statistical test of WPLI between the experimental and the control groups during the aiming period, some statistically significant connections and nodes were found in the three frequency bands. According to the experimental results, the shooter has higher functional connection strength in alpha and beta frequency bands during aiming under noise conditions. The alpha band showed significant differences in the left and right temporal lobes and sensorimotor cortex, while the beta band showed significant differences in the right frontal region, the left parietal region, and the sensorimotor cortex. The temporal region is the auditory center, which is responsible for the sense of sound. The right frontal region is related to social consciousness, while the left parietal region is related to mathematics, problem-solving, and complex grammar [47]. Alpha rhythm is related to attention, lucid thinking, and integration. It plays an important role in the integration of brain structure in sensory and cognitive activities and mainly occurs in brain areas that do not participate in work [48,49]. Combined with the characteristics of alpha and beta rhythms’ inhibition and activation of coping with specific brain regions, noise disturbance was speculated to cause shooters to activate more brain regions’ information interaction during aiming, but some brain regions unrelated to shooting behavior are also awakened, and more energy is mobilized to suppress the disturbance caused by noise.

Under the weak-light condition, the left frontal region of the alpha band weak-light group was lower than that of the control group, while the right occipital region of the beta band was significantly higher than that of the control group. The left frontal area is related to executive planning, working memory, and concentration. The occipital lobe is the center of the visual cortex [47]. The weak-light group showed lower information interaction in the left frontal region, which means that the inhibition of related functions in the brain region was weaker, and the occipital cortex was significantly activated. Previous studies have found that short-term visual deprivation can increase the excitability of the occipital visual cortex [25,26], which is also consistent with the conclusion of this paper. Therefore, the occipital cortex of the brain is believed to be more significantly awakened, and the communication of visual information is closer when the shooter aims under weak-light conditions.

### 4.2. Analysis of the Dynamic Change of Function Connection in the Shooting Preparation Stage

By evaluating the change of mean WPLI value of the shooter during the entire shooting process between the baseline and aiming periods, a significant difference was only found between the baseline and aiming periods in the theta band. The experimental results show that the entire brain has a higher connection value in the baseline period in this frequency band. Theta rhythm is usually closely related to memory coding and retrieval, working memory retention, and the need to achieve top-down control in high-level cognitive processes [50,51,52]. Previous studies have found a decrease in theta rhythm in subjects with increased attention and alertness [50,53,54]. This indicates that as the shooting time approaches, the brain’s information communication in theta band will gradually weaken, which may also mean that the shooter’s attention becomes continuously focused during aiming.

ERWPLI reflects changes in the brain’s functional connectivity strength during aiming related to baseline. Further analyzing the change process of the neural mechanism in the shooting preparation stage is helpful by comparing the ERWPLI in different shooting conditions. In the comparison between the noise and the control groups, the control group showed a higher change rate of functional connectivity in the right frontal, right temporal, and right parietal regions in the beta band. According to the results of the nodes with significant differences in ERWPLI in the midbrain topographic map, the information interaction ability of the right brain region in the beta band is believed to be lower than that in the normal condition during the entire shooting process of the shooter under the noise disturbance. The right hemisphere is related to spatial awareness, target recognition, and visual memory [47,48]. This seems to indicate that the skilled shooter’s cognitive processing ability of target recognition, working memory, and other motion-related functions during the entire shooting process under the noise disturbance is relatively weaker than that in the normal condition.

In weak-light conditions, the ERWPLI of the shooter in the alpha band was significantly higher than that of the control group, and these significant nodes were mainly concentrated in the left temporal region. In the weak-light environment, the shooter strengthened the information exchange near the left parietal region during aiming. According to the physiological significance of the left parietal region of the brain described above, skilled shooters may have more significantly inhibited the information processing of the left parietal region that has nothing to do with shooting behavior in a weak-light condition.

Combined with the analysis conclusion of functional connectivity differences during aiming in Section 4.1, the shooters may show relatively focused attention under the condition of restricted audiovisual senses. They enhanced the arousal of more cerebral cortex under the noise disturbance condition, which helped to suppress the interference caused by noise, but also caused an increase in the mental load of the shooter, and the information interaction ability between brain regions corresponding to motor behavior was weaker than that in the normal condition. In the weak-light condition, shooters can focus more on the processing of visual information by inhibiting the information interaction in unrelated brain regions.

### 4.3. Analysis of Differences in Brain Network Topology during Aiming

Brain network analysis based on graph theory can characterize the connection mode within the brain from the perspective of the topological organization to effectively evaluate the topological structure of the functional network of the brain [55]. The difference in the global and local brain network topology of the shooter during aiming under three conditions was tested. According to the results of global topological characteristics, the noise group shows stronger global efficiency, average clustering coefficient, and smaller characteristic path length than the other two groups in the beta band. In the alpha band, the same results were found in the noise and weak-light groups. Global efficiency can measure the global transmission capacity of the network and the efficiency of information integration [56]. The clustering coefficient indicates the closeness of nodes and neighbor nodes in the network [57]. The characteristic path length is defined as the average of the shortest lengths of the paths between any two nodes in the network and measures the degree of the overall routing efficiency of the network [58]. They all reflect the global information conversion efficiency of the brain network [59]. This shows that the structure of the shooter’s brain network is more optimized during aiming under noisy conditions, and it has stronger overall routing efficiency and information processing capabilities.

In addition, the average clustering coefficient and characteristic path length are the measurement indicators that reflect the functional separation and integration in the functional network of the brain [60], and their ratios reflect the small world of the network [61]. Previous studies believe that the small world feature helps to optimize information transmission in the network, improve learning efficiency, and support separation and distributed information processing [62,63]. Compared with the weak-light condition, the shooter’s higher average clustering coefficient and lower characteristic path length under the noise disturbance condition indicate that the shooter’s brain network in this condition has the characteristics of a small world; the overall information exchange of the brain network is closer, the organizational efficiency is higher, and more clustering phenomenon is observed.

The analysis of local topology characteristics helps understand the brain regions with differences between brain networks during aiming under different sensory constraints. From the significant difference in local efficiency reflected in the brain map, the noise group showed higher local efficiency than the control group in the three frequency bands. The local efficiency of the weak-light group was lower than that of the control group only in the left frontal region in the alpha band. The local efficiency of the brain network reflects the information transmission efficiency of this node in the network [38]. This shows that the shooter under noise disturbance conditions has higher global efficiency during aiming and also shows stronger information transmission and processing efficiency than in the normal condition in the left frontal region in the theta band, the left and right temporal and parietal regions in the alpha band, and most regions of the entire brain in the beta band. The efficiency of information exchange in the shooter’s left frontal area during aiming in the weak-light condition is lower than that in normal conditions, which is also consistent with the nodes with significant differences in functional connections during aiming.

A noticeable phenomenon was found in the analysis of the eigenvector centrality. In the comparison results in the three frequency bands, the eigenvector centrality of the left temporal region in the theta band and the right temporal region in the beta band was lower in the noise group than in the control group. Under the weak-light condition, the eigenvector centrality during the aiming of the shooter in the three frequency bands is significantly lower than that in the normal condition. The eigenvector centrality of a node indicates the importance of the node in the network [43]. The temporal and occipital regions are involved in the processing of auditory and visual information, respectively. The eigenvector centrality in these two brain regions, which are highly related to audiovisual function, is significantly lower than that in the normal condition. This may indicate that shooting under sensory-restricted conditions will cause the brain to reduce the importance of information interaction in these brain regions related to restricted sensory function for the functional network in the entire cerebral cortex. This seems to be a meaningful discovery and deserves further study.

### 4.4. Correlation Analysis between Brain Network Characteristics and Firing Performance in the Shooting Preparation Stage

This study examines differences while also analyzing correlations. It is hoped that by comparing the connections and nodes with a significant correlation between brain network characteristics and shooting performance in different environments, the impact of different conditions on neural activity in the shooting process can be comprehensively analyzed. According to the results of the Spearman rank test, on the whole, in addition to the eigenvector centrality, more connections exist between the weak-light group and the shooting performance in other brain network characteristics than the noise group. This also corresponds to the shooting performance of the weak-light group, which is significantly lower than that of the other two groups. It indicates that a certain degree of noise will not have a serious impact on the shooter’s cognitive function, so the shooter’s shooting performance in this condition is more independent, and the weak-light condition has a greater impact on the shooting performance.

Most of the significant connections in the theta band of the noise and weak-light groups were found to be negatively correlated during aiming, while more significant connections in the alpha and beta frequency bands were positively correlated. This seems to suggest that the more efficient pairing of information in the alpha and beta bands contributes to improved shooting performance, whereas the theta band does the opposite. Meanwhile, in the alpha band, the functional connections of skilled shooters in two sensory-restricted conditions are positively correlated with their shooting performance, and the connections are mostly concentrated on the nodes in the left hemisphere. Similarly, the change rate of functional connectivity in both the noise and weak-light groups showed more positive and negative connections in the left and right hemispheres, respectively, in the alpha band. The left hemisphere is in charge of language understanding, logical thinking, analysis, and calculation [47]. Some previous studies on brain power during shooting and aiming have also found similar phenomena [5,64,65]. Combined with the conclusions of these studies, the phenomenon of the left hemispheric laterality in this correlation can be explained as the increase in the left hemispheric alpha frequency band during the entire shooting process that helps to inhibit the information interaction in the cerebral cortex unrelated to the shooting behavior, which helps to further awaken the brain regions related to the shooting behavior to improve the shooting performance.

## 5. Limitation

This study is a preliminary attempt to study the effect of audiovisual sensory limitation on the cognitive ability of skilled shooters in the shooting preparation stage through physiological measurement. Although the methods of brain connectivity and brain network topology are used for comprehensive analysis, a scientific and accurate explanation is still lacking for all the significant complex connections and the significant positive and significant negative correlations in the experimental results. This may also be because the number of subjects is limited and the composition of the sample is single, so the conclusion of the study is not universal. Another limitation is that compared with a real complex environment (e.g., a law enforcement site), the subjects in this study have relatively short exposure time to noise and weak light, and the noise type is continuous rather than intermittent, resulting in less impact than the actual situation. Meanwhile, the experiment only studied the influence of noise disturbance of 70 dB and weak-light condition of 25 lx on shooting and also lacked a comparison with EEG characteristics in the shooting preparation stage under different noise volumes and light intensities, which made the conclusion of this study only applicable to limited scenes. In addition, the WPLI used in the experiment as a method to analyze brain connectivity and construct a functional connectivity matrix solves the common source problem to a certain extent, but the brain network matrix obtained does not have directionality. The brain network parameters selected in this study are relatively few. In future research, it can be further expanded to add network parameters based on graph theory, such as rich club coefficient, modularity, transitivity, etc., and use methods such as multidimensional array (tensor), multi-modality, depth learning, etc., for comprehensive analysis, so as to promote the study of brain network characteristics in the process of shooting related sports.

## 6. Conclusions

This study analyzed the difference in brain network characteristics of 30 skilled shooters in the shooting preparation stage under noise disturbance and weak-light conditions and the correlation with shooting performance. The shooter’s mean WPLI value in the theta band was found to significantly decrease in the three conditions as the shooting time approached, suggesting that the shooter’s attention continuously improved. A noise of 70 dB mildly interferes with the shooter’s cognitive ability during aiming. In the noise disturbance condition, the overall wake-up degree and routing efficiency of the brain were higher, and the topology of the brain network was more optimized; however, it increased the mental load of the shooter. The weak-light environment had a greater impact on the shooting performance. It made the shooter concentrate more on visual information processing during aiming and strengthened the functional inhibition of brain regions unrelated to shooting behavior. Additionally, during the eigenvector centrality analysis, audiovisual interference was found to make the cortical region corresponding to the audiovisual sensory function in the shooter’s brain less important in the entire brain network than in the normal condition. These conclusions verify the hypothesis. Therefore, providing a valuable reference for functional connectivity and complex brain network studies is projected when shooters perform shooting-related motor tasks under limited conditions of multisensory function.

## Figures and Tables

**Figure 1 brainsci-12-01373-f001:**
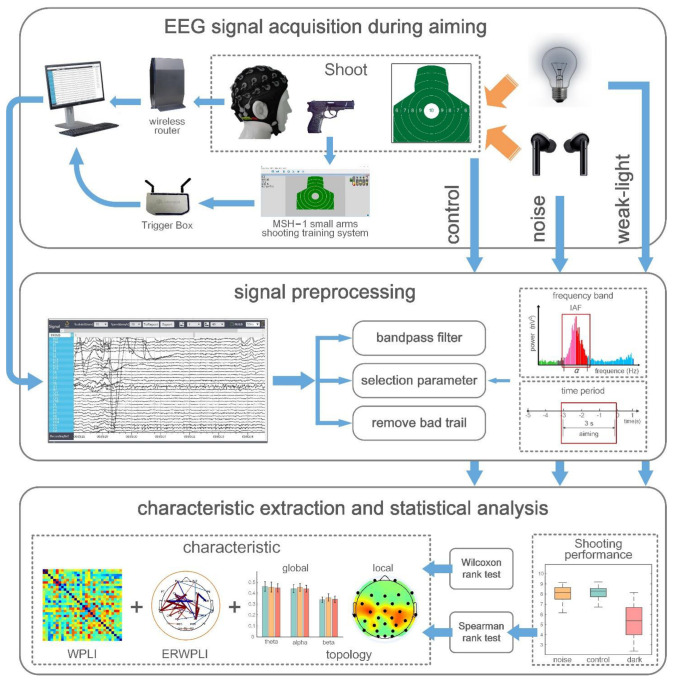
The experimental analysis process.

**Figure 2 brainsci-12-01373-f002:**
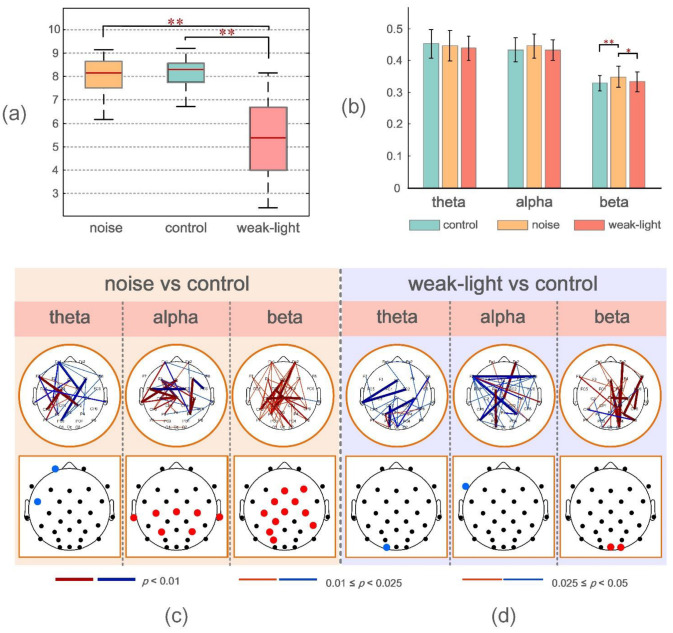
(**a**) From left to right, the mean, the standard deviation of the mean scores of the noise group, the control group, and the weak-light group. * *p* < 0.05 indicates a significant difference in topological characteristics; ** *p* < 0.01 indicates a very significant difference in topological characteristics. (**b**) Mean and standard deviation of the mean WPLI values during shooting aiming at theta, alpha, and beta frequency bands in the control, noise, and weak-light groups. The vertical axis in the figure represents the functional connectivity strength, and the error bars represent the standard deviation. (**c**,**d**) represent the connection and electrode positions where the WPLI connection values of each channel and the average WPLI connection values of each node in the noise and weak-light groups are significantly different from those in the theta, alpha, and beta bands of the control group, respectively. The black nodes in the figure represent the electrode positions. The colored connections and nodes indicate a statistically significant (*p* < 0.05) difference between the connection in the different leads. Red/blue in the figure represent the connections and nodes in the experimental group (noise group, weak-light group) with higher/lower connection values than those in the control group. The color bar indicates statistically significant differences in connection values.

**Figure 3 brainsci-12-01373-f003:**
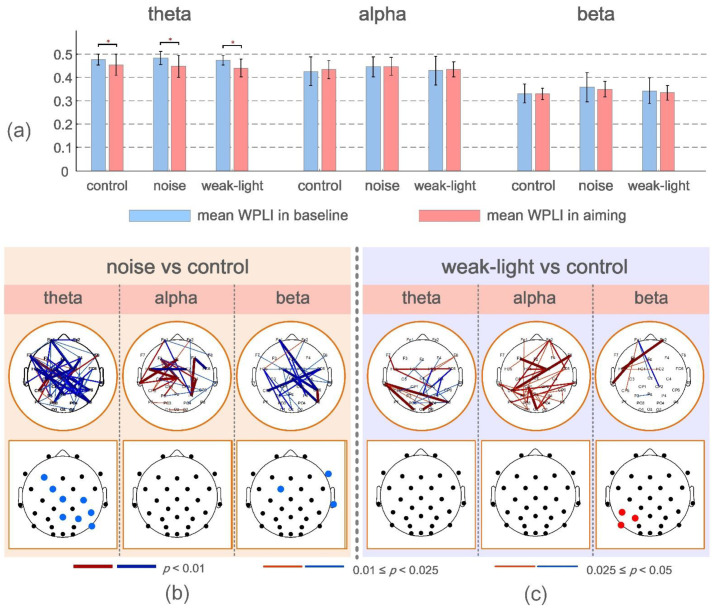
(**a**) Mean and standard deviation of the mean WPLI values at baseline and during aiming for the three conditions. * *p* < 0.05 indicates a significant difference. (**b**,**c**) Connections and nodes with significant differences in ERWPLI in theta, alpha, and beta bands between the noise, the weak-light, and the control groups. Red/blue indicates that the ERWPLI of the noise and the weak-light groups was higher/lower than that of the control group.

**Figure 4 brainsci-12-01373-f004:**
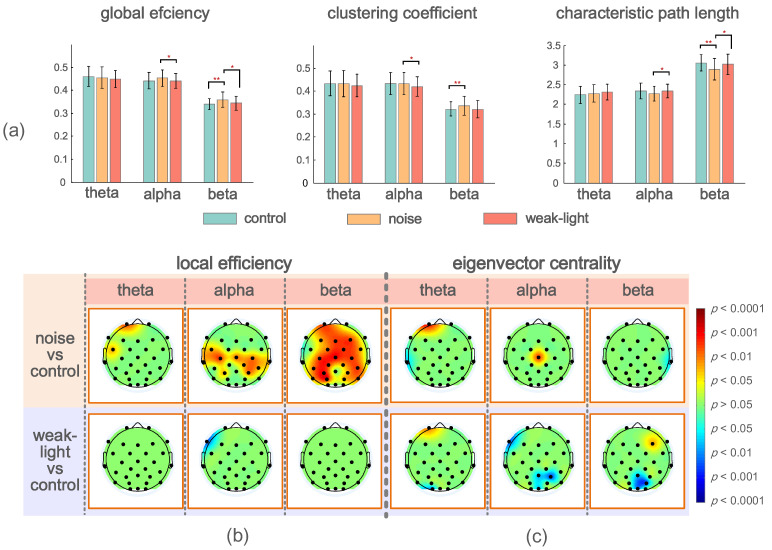
(**a**) Mean and standard deviation of global topological characteristics of brain network in theta, alpha, and beta bands of the control, noise, and weak-light groups. * *p* < 0.05 indicates a significant difference in the topological characteristics, and ** *p* < 0.01 indicates a very significant difference in the topological characteristics. (**b**) Topographic maps show significant local efficiency differences between the noise and weak-light groups and the control group. Red and blue are the electrode positions whose characteristic values were significantly higher/lower than those of the control group. (**c**) The topographic map shows significant differences between the eigenvector centrality in the noise, weak-light, and control groups.

**Figure 5 brainsci-12-01373-f005:**
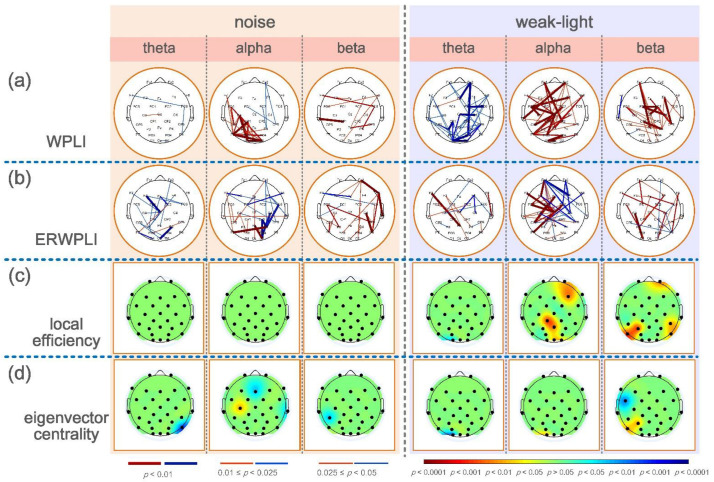
(**a**,**b**) WPLI and ERWPLI of skilled shooters in the theta, alpha, and beta bands are significantly correlated with shooting performance in the noise and weak-light conditions (*p* < 0.05). Red and blue shows positive (*r* > 0) and negative (*r* < 0) correlations. (**c**,**d**) Brain topographic maps showing the correlation between local efficiency, eigenvector centrality, and shooting performance. Red/blue is the electrode position with a significant positive/negative correlation with shooting performance.

## Data Availability

The data that support the findings of this study are available from the corresponding author upon reasonable request.

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
