# Peer review of "Brain Network Research of Skilled Shooters in the Shooting Preparation Stage under the Condition of Limited Sensory Function"

_brainsci, 2022, doi:10.3390/brainsci12101373_

Round 1

Reviewer 1 Report

The article discusses the neurological effects of distractions, mainly noise and low light, on the performance of a shooter. The effects of distractions are studied using EEG measurements. The article discusses an important issue as understanding the cognitive and psychological behavior of the armed force personnel in an environment with distractions can assist in better and more efficient training. The article is very well written with an adequate explanation of the experiment design, analysis techniques, and discussion of the results. I do have some minor comments about grammatical errors and formatting issues here and there.

1.       Missing reference on line 198

2.        I believe line 331 should be ‘However, it was insignificant in the . . .’  and not ‘significant’.

3.       Line 347 mentions significant difference in the alpha band but figure 3c suggests it's in beta band. Please correct that.

4.       Lines 351 to 353 seem to be incomplete. I believe something is missing in between.

5.       Line 361 doesn’t make sense. ‘control group than that of the control group’? I believe the author meant noise group vs control group.

6.       Please change the autocomplete © to ( c ) in line 419.

7.       Line 429 mentions figure 6 (d) but there is no figure 6 in the manuscript. I guess the author meant figure 5 (d).

8.       Figure caption for figure 5 mentions ‘archery performance’ on line 414. I believe it should be shooter performance. Archery performance was the previous study. Make sure to make relevant changes when you copy stuff from old manuscripts. 

Author Response

We sincerely show the gratitude to the Editor and Reviewers for their careful and constructive comments on the manuscript. We have addressed all the comments and made changes as appropriate. The revised parts of the paper are marked in yellow.

Question 1:Missing reference on line 198

Answer: Thanks for your advice. We have supplemented the citations in this part, on page 5, line 188.

Q2:I believe line 331 should be ‘However, it was insignificant in the . . .’  and not ‘significant’.

Answer: Thank you for pointing out the shortcomings of our paper. We have modified it, on page 10, line 322.

Q3: Line 347 mentions significant difference in the alpha band but figure 3c suggests it's in beta band. Please correct that.

Answer: Thanks for your careful and comprehensive review of this manuscript. This is because we mistakenly reversed the alpha band diagram with the beta band diagram when making the diagram, which has been corrected in Figure 3.

Q4: Lines 351 to 353 seem to be incomplete. I believe something is missing in between.

Answer: Thanks for your careful and comprehensive review of this manuscript. We have modified the content of this section, which is on page 12, lines 342 to 343.

Q5: Line 361 doesn’t make sense. ‘control group than that of the control group’? I believe the author meant noise group vs control group.

Answer: Thanks for your careful and comprehensive review of this manuscript. We mistakenly wrote the experimental group (i.e. weak-light group and noise group) as the control group, and we modified the contents of this part. On page 12, line 352.

Q6: Please change the autocomplete © to ( c ) in line 419.

Answer: Thank you for your advice. We have modified the content of this part, which is on page 15, line 410.

Q7: Line 429 mentions figure 6 (d) but there is no figure 6 in the manuscript. I guess the author meant figure 5 (d).

Answer: Thanks for your careful and comprehensive review of this manuscript. We have modified the content of this part, which is on page 15, line 420.

Q8: Figure caption for figure 5 mentions ‘archery performance’ on line 414. I believe it should be shooter performance. Archery performance was the previous study. Make sure to make relevant changes when you copy stuff from old manuscripts.

Answer: Thank you for your advice. We have changed this to "shooting performance" in line 405 on page 15.

Finally, thank you again for your careful review of our manuscript.

Reviewer 2 Report

"Brain network research of skilled shooters in the shooting preparation stage under the condition of limited sensory function" is an interesting article. The aim was to analyze the functional connectivity and brain network characteristics of marksmen in the preparation stage of pistol shooting under noise disturbance and weak-light conditions by simulating an environment with limited auditory and visual functions. The results shown that noise disturbance activates the arousal level of the shooter’s brain and enhances the information processing efficiency of the brain network. In weak-light conditions shooters focus more on visual information processing during aiming and strengthen the inhibition of functions in the brain regions unrelated to shooting behavior. Audiovisual disturbance renders the cortical regions equivalent to the audiovisual perception function in the shooter’s brain less important in the entire brain network than the normal condition.

There are some issues in the article that need to be addressed.

Introduction

Line 43 and 44. The term psychological factors should be used with caution since it is confusing, it is not clear what exactly it refers to, and nowhere in the introduction is it used or defined adequately. It is suggested to omit this term or define it.  

Line 109. It is implied that these sentences describe the objective of the study. If so, it is suggested to make it very clear in the wording. 

Lines 117 to 126. In my opinion this paragraph is out of place, it corresponds rather to the methodology used in the study and should be moved or integrated in the corresponding section: Materials and Methods. 

Lines 125 and 126. A fundamental part of the article refers to the use of neural network analysis using graph theory. I think that this item should be mentioned and described more fully in the Introduction. 

In general, the Introduction is adequate, but it is suggested to carefully review the wording since it is sometimes redundant and perhaps a bit excessive. This would improve the clarity of the revised concepts. 

Material and Methods 

Line 134. It should be added if the experiments were performed at the same time, its relationship with the last food intake and how the amount of sleep was controlled one night before the experiment. These factors may influence the results. 

Line 167. It is suggested to omit Figure 1. The description of the derivations used is sufficient and the figure is well known to readers. It is suggested to add the gain of the channels used, and the low and high filters used during the EEG recording. 

Line 170. It must be justified and described why resting EEG was used with closed eyes and why not with open eyes, since the experimental comparisons are with open eyes. 

Lines 214 and 215. It is suggested to add the meaning of WPLI. 

Results 

Line 338. The meaning of ERWPLI should be described. 

Discussion 

Line 485. Again the term “psychological state” is used, but in the following sentences only cognitive processes are described. It is suggested to avoid the term. 

Line 487. The influence of the baseline being done with eyes closed should be discussed. 

To complete the discussion, it is suggested to analyze the impact that the results obtained have on the neural networks and the theta, alpha and beta frequency bands, specifically on the processes of attention and visuospatial working memory that are involved in shooting performance.

References

A careful review of all references is suggested. Some references have the names of the Journals in uppercase and others in lowercase. There is inconsistency in the way of citing, on some occasions the volume of the Journal is given and on others not, even from the same journal (Neuroimage). Some citations do not have pages (Reference 11) 

Author Response

We sincerely show the gratitude to the Editor and Reviewers for their careful and constructive comments on the manuscript. We have addressed all the comments and made changes as appropriate. The revised parts of the paper are marked in yellow.

Question 1:Line 43 and 44. The term psychological factors should be used with caution since it is confusing, it is not clear what exactly it refers to, and nowhere in the introduction is it used or defined adequately. It is suggested to omit this term or define it.

Answer: Thanks for your advice. We have deleted the statement in this part.

Q2:Line 109. It is implied that these sentences describe the objective of the study. If so, it is suggested to make it very clear in the wording.

Answer: Thank you for pointing out the shortcomings of our paper. We rewrote this part to highlight the research purpose. See page 110-112 on page 3 for details.

Q3: Lines 117 to 126. In my opinion this paragraph is out of place, it corresponds rather to the methodology used in the study and should be moved or integrated in the corresponding section: Materials and Methods.

Answer: Thanks for your advice. According to your suggestion, we have deleted the statement in this part to make the introduction more concise.

Q4: Lines 125 and 126. A fundamental part of the article refers to the use of neural network analysis using graph theory. I think that this item should be mentioned and described more fully in the Introduction.

Answer: Thanks for your careful and comprehensive review of this manuscript. According to your suggestion, we have added more expressions about brain network related knowledge based on graph theory in the introduction. Since this part of the original text has been deleted, we put the new part on page 3, lines 99-103.

Q5: Line 134. It should be added if the experiments were performed at the same time, its relationship with the last food intake and how the amount of sleep was controlled one night before the experiment. These factors may influence the results.

Answer: Thank you for your advice. The subjects in the experiment were all students of the Armed Police Engineering University. They were managed in a closed way in the school on weekdays and kept a regular schedule consistent with that in the army, so they all kept enough sleep. Similarly, his diet was the same as that in the army. It was light on the whole, and he did not eat any stimulating food that would affect his mental state. We added the corresponding statement on page 4, lines 152-126.

Q6: Line 167. It is suggested to omit Figure 1. The description of the derivations used is sufficient and the figure is well known to readers. It is suggested to add the gain of the channels used, and the low and high filters used during the EEG recording.

Answer: Thank you for your advice. According to your suggestion, we deleted Figure 1. The common mode rejection ratio of the EEG amplifier used in the experiment and the high-low pass filter used in the recording are added in lines 151-153 on page 4.

Q7: Line 170. It must be justified and described why resting EEG was used with closed eyes and why not with open eyes, since the experimental comparisons are with open eyes.

Answer: Thanks for your careful and comprehensive review of this manuscript. First of all, the baseline value of our manuscript is - 4 to - 3s EEG before shooting during aiming, rather than resting EEG. The EEG signal in eyes closed resting state is used because the individual difference is considered in the frequency division of manuscripts, and the frequency division based on IAF is adopted. The EEG signal of the occipital region in the closed eye resting state is only used in this part of the full text, because Klimesch proposed in 1996 that the peak alpha power of each subject in the closed eye resting state is taken as the IAF of each subject (Klimesch, W., 1996. Memory processes, brain ossifications and EEG synchronization. International Journal of Psychology 24 (1 – 2), 61 – 100 Review.). Therefore, on the basis of previous studies, this study also selected the alpha power of eyes closed resting state as the IAF of each subject. Relevant references are added to the corresponding positions in the manuscript.

Of course, we also found that the expression of IAF in the manuscript was not translated accurately enough, so we revised the expression of this part. For details, see lines 480-190 on page 5 of the manuscript.

Q8: 214 and 215. It is suggested to add the meaning of WPLI.

Answer: Thanks for your advice. We added the corresponding content in lines 204-205 on page 6.

Q9: The meaning of ERWPLI should be described.

Answer: Thanks for your advice. According to your suggestion, we added the meaning of ERWPLI in line 329 on page 11 of the results section. At the same time, we have made some amendments to the ERWPLI section in Materials and Methods, which is shown in lines 222-224 on page 7.

Q10: Line 485. Again the term “psychological state” is used, but in the following sentences only cognitive processes are described. It is suggested to avoid the term.

Answer: Thanks for your careful and comprehensive review of this manuscript. We have deleted the statement in this part.

Q11: The influence of the baseline being done with eyes closed should be discussed.

Answer: Thanks for your advice. As we all know, there is a great difference between the nerve state of shooters in shooting tasks and that in resting states. This part of this study aims to explore the change of functional connection strength during shooting aiming under different jamming environments, not to explore the difference between visual aiming period and resting state. Therefore, in the experiment, we choose - 4s~- 3s before shooting during the shooter's aiming as the baseline period. As mentioned earlier, the purpose of collecting EEG signal of eye closed resting state is to divide the frequency band according to IAF (alpha power peak of eye closed occipital region).

Q12: To complete the discussion, it is suggested to analyze the impact that the results obtained have on the neural networks and the theta, alpha and beta frequency bands, specifically on the processes of attention and visuospatial working memory that are involved in shooting performance.

Answer: Thanks for your advice. In the discussion part, we mainly analyze and explain the research results based on the physiological characteristics corresponding to the frequency band, focusing on establishing the relationship between the research results and shooting behavior. For example, No. In the analysis, because there are more significant results in the alpha and beta bands, and more research has been done on these two bands during the shooting aiming period, we focus on the results in the alpha and beta bands.

We attach great importance to your comments and regard them as the direction of our next efforts. Unfortunately, we are researchers in the field of communication, so we pay more attention to the research of signal analysis and feature extraction methods in the paper. Maybe the research on cognitive theory and biological anatomy is not enough, which is also the main aspect that we need to improve in the next step.

Q13: A careful review of all references is suggested. Some references have the names of the Journals in uppercase and others in lowercase. There is inconsistency in the way of citing, on some occasions the volume of the Journal is given and on others not, even from the same journal (Neuroimage). Some citations do not have pages (Reference 11)

Answer: In the reference section, we have modified the format of all references one by one. For document 11 you mentioned, we have tried many methods to find the page of the document. Of course, we have provided you with references to the document in other articles: Zhang, J., Shi, Y. X., Wang, C. K., Cao, C., Zhang, C., Ji, L., et al. (2021) Pre-shooting electroencephalographic activity of professional shooters in a competitive state. Comput. Intell. Neurosci. 2021:6639865. doi: 10.1155/2021/ 6639865.

Through the revision of the paper, we have further understandings of the relevant knowledge, and also understand that there are many shortcomings in my research in the relevant field and need to be improved. Your final opinion is very important to us. If we still need to modify the manuscript, we will respect your opinion and resolutely correct it. Finally, once again sincerely thank you for your suggestions on our paper.

Reviewer 3 Report

I consider the study well designed, with an interesting proposal regarding the subject and the analysis, despite the limitations of the technique and the sample, which could be considered with more emphasis in the discussion.

Author Response

We sincerely show the gratitude to the Editor and Reviewers for their careful and constructive comments on the manuscript. We have addressed all the comments and made changes as appropriate. The revised parts of the paper are marked in yellow.

Question 1:I consider the study well designed, with an interesting proposal regarding the subject and the analysis, despite the limitations of the technique and the sample, which could be considered with more emphasis in the discussion.

Answer: Thanks for your advice. Thank you for your careful review and approval of our manuscript. According to your suggestion, we have deepened the discussion on the limitations of the study in terms of technology and samples. The specific modifications are in the Limitation section on page 19.
